# The Influence of *Lactiplantibacillus plantarum* and *Oenococcus oeni* Starters on the Volatile and Sensory Properties of Black Raspberry Wine

**DOI:** 10.3390/foods12234212

**Published:** 2023-11-22

**Authors:** Changsen Wang, Shuyang Sun, Haoran Zhou, Zhenzhen Cheng

**Affiliations:** 1School of Food Engineering, Ludong University, Yantai 264025, China; wangchangsenau@126.com (C.W.); zhrxjz@sina.com (H.Z.); zhenzhencheng30@163.com (Z.C.); 2College of Food Science & Nutritional Engineering, China Agricultural University, Beijing 100083, China

**Keywords:** black raspberry wine, *Lactiplantibacillus*, *Oenococcus*, GC-IMS, sensory evaluation

## Abstract

Malolactic fermentation (MLF) by different lactic acid bacteria has a significantly influence on the aromatic and sensory properties of wines. In this study, four strains including two *Oenococcus oeni* (commercial O-Mega and native DS04) and two *Lactiplantibacillus plantarum* (commercial NoVA and native NV27) were tested for their performances over MLF and effects on the basic composition, volatile components and sensory property of black raspberry wine. Results of microbial growth kinetics showed *Lactiplantibacillus* strains had higher fermentation efficiency than *Oenococcus*. The volatile compounds were determined by GC-IMS; NoVA and NV27 had higher production of volatile esters, and DS04 synthesized more amounts of acetate esters and several alcohols. In terms of sensory evaluation, NV27 and DS04 showed great aroma properties due to the enhanced fruity and sweet aroma. Furthermore, PLS was used for the establishment of the relationship between volatiles and sensory odors and sensory data interpretation.

## 1. Introduction

The participation of fermentation microorganisms is essential for the production of fruit wines. Fruit wine fermentation usually consists of two stages, i.e., alcoholic fermentation (AF) and malolactic fermentation (MLF). MLF is a secondary fermentation process that usually takes place during or after the completion of alcoholic fermentation, and is triggered by the activity of lactic acid bacteria (LAB). And such a process had been reported to contribute to the wine’s acidity reduction, stability improvement, flavor enhancement and palatability improvement [1,2,3]. Nowadays, *Oenococcus oeni* and *Lactiplantibacillus plantarum* are the two most common bacterial species used in MLF. *L. plantarum* is excellent in the formation of a more complex sensory profile and prevention of several undesired compounds than *O. oeni* [4,5,6]. However, *O. oeni* is favored over *L. plantarum* as an industrial starter in MLF due to its more efficient malate metabolism performance under harsh wine-making conditions [7,8]. Compared to spontaneous MLF, controlled fermentation with commercial LAB starters has the advantage of assuring a rapid and reliable fermentation process and offering the wine a consistent quality [9]. However, highly strain-dependent variability of the same LAB species also leads to the divergences of wine chemical and aroma quality, which further increased the difficulties of selecting appropriate MLF starters. To solve such problems, inoculating indigenous or locally selected wine LAB strains with preferable oenological characteristics is encouraged, due to the advantages they possessed, such as better environment adaptability and more enhancement of wine’s characteristics.

Black raspberries have been a staple food for hundreds of years, and are popular for their taste and the abundance bioactive substances [10]. However, the preservation of mature black raspberries is not satisfactory; thus, it is often processed to juice and wine. Nowadays, the consumption of raspberry wine has become more popular in China, and it is out of question that the special characteristic and distinct flavor of this product will receive more scrutiny in the future. Aroma is one of the main characteristics that determine black raspberry wines’ organoleptic quality and style, and wine-associated microorganisms seem to play a significant role in its formation and modulation. Up to now, only limited studies have been addressed upon the microbial effect on the fermentation and quality of such a wine product. Researchers used gas chromatography and mass spectrometry to analyze volatile flavor compounds produced by fermentation of black raspberry wine with traditional yeasts [11]. Head-space gas chromatography-ion migration spectrometry (HS-GC-IMS) was used to investigate how wine-related microbial interactions alter sensory properties and quality of black raspberry wine [12]. They also investigated the effects of a mixed inoculation on black raspberry wine [13].

Based on the fact that there is limited research on the influence of LAB starters on the sensory property and quality of black raspberry wines, it enables us to initiate studies around such subjects. In our previous study, two indigenous LAB isolated from the local black raspberry orchard, identified with preferable technological properties including high fermentation speed and high tolerance to wine’s harsh conditions, were tested for their fermentation characteristics in actual black raspberry wine fermentation. Also, with the aim of making comparisons of the difference between *L. plantarum* and *O. oeni*, two commercial LAB strains were used in the parallel experiments. During fermentation, the inoculum size of LAB was determined, and after MLF resultant black raspberry wines were analyzed and compared for analytical, aromatic and sensory profiles. The obtained results are expected to enrich our understanding on the effects of local MLF strains on the aroma composition and sensory quality of black raspberry wine.

## 2. Materials and Methods

### 2.1. Black Raspberries and Wine-Making-Related Microorganisms

Black raspberries of “Hull”, picked at commercial maturity from the local orchard (Yantai, China) during the 2020 harvest season were used in this work. The analytical data of the initial musts were: total sugars 110.4 g/L, total acidity 9.7 g/L, pH 3.42.

Three commercial strains including *S. cerevisiae* of Lalvin D21 (Lallemand Inc., Montreal, QC, Canada), *O. oeni* of O-Mega (Chr.-Han, Horsholm, Denmark) and *L. plantarum* of Viniflora NoVA (Chr.-Han, Horsholm, Denmark) were used for the fermentations. And two native strains, an *O. oeni* of DS04 and *L. plantarum* of NV27, isolated from black raspberry wines in Yantai region (China) with appropriate characteristics for wine production, were also tested in this study.

### 2.2. Production of Black Raspberry Wine

Black raspberries were crushed and then treated with 20 mg/L of Lallzyme EX-V pectinase and 50 mg/L of potassium metabisulphite, and sucrose was supplemented to obtain a total reducing sugar content of 210 g/L to ensure that its AF can meet the alcohol requirements of commercial black raspberry wine. Alcoholic fermentation was started by the introduction of D21 at a dosage of 250 mg/L, and carried out at 24 °C under static condition till the sugar concentration decreased below 4 g/L. When alcoholic fermentation was finished, the yeasts were immediately removed by centrifugation at 7000 rpm for 20 min, and the resultant black raspberry wines were divided into twelve 10 L vessels for the subsequent MLF. Four MLF starters, including *O. oeni* O-Mega, *O. oeni* DS04, *L. plantarum* Viniflora NoVA and *L. plantarum* NV27, were tested in this study, and for simplicity, we defined BRO1, BRO2, BRL1 and BRL2 as the black raspberry wines fermented by the four LABs, respectively.

MLF was started by the inoculation of selected LAB at a concentration of 10^6^ cfu/mL. Before inoculation, the LAB were precultured in MRS broth at 30 °C for 24 h, and the biomass was centrifuged at 8000 rpm for 10 min and resuspended in water. And MLF was carried out at 20 °C under static condition. L-malic acid content was monitored by Enzymatic BioAnalysis (Boehringer-Mannheim/R-Biopharm, Darmstadt, Germany) until its concentration was reduced below 0.1 g/L. When MLF was completed, all the young raspberry wines were centrifuged, sulphited to 50 mg/L of free SO_2_ and stored at 4 °C until the analysis.

### 2.3. Cell Growth during Fermentation

The growth and survival of microbes were measured by plate counting. Samples were extracted through periodic fermentation and properly diluted with 0.85% (*w*/*v*) sterile NaCl solution. *O. oeni* was identified and counted according to colony morphology after 3 d of 28 °C incubation using Wallerstein Laboratory Nutrition (WLN). After 6 d of 28 °C incubation, *O. oeni* was calculated on the ATB AGAR plates. To prevent bacteria from growing, 50 mg/L of cycloheximide (Wako Pure Chemical Industries Inc., Osaka, Japan) was added to AGAR ATB. For *L. plantarum* counts, MRS agar fresh plates (Scharlau Chemie S.A., Barcelona, Spain) with 0.2 mg/mL of nystatin (Acofarma, S. Coop., Terrassa, Spain) were incubated at 30 °C under strict anaerobic conditions for at least 5 days. Viable counts were recorded as the number of cfu/mL.

### 2.4. Basic Analysis

Conventional oenological parameters, including total reducing sugars (Lane–Eynon method, g/L), total acidity (potentiometric titration, calculated as g/L of malic acid), pH (Sartorius PB-10, Göttingen, Germany), ethanol (distillation, %), volatile acidity (distillation using a Cazenave–Ferré method followed by titration with phenolphthalein, expressed as g/L of acetic acid) and dry extract (density bottle method, g/L), were determined according to the OIV official methods [14].

### 2.5. Volatile Fingerprint Analysis

The determination of volatile components in black raspberry wine was conducted gas on a chromatography-ion migration spectrometry (GC-IMS) (FlavourSpec^®^, Gesells-chaft für Analytische Sensorsysteme mbH, Dortmund, Germany) with a WAX column (30 m × 0.53 mm i. d., 0.1 μm film thickness), methods and parameters refer to and improve upon existing research [12]. Briefly, 1 mL wine sample was placed in a 20 mL headspace bottle, incubated in the automatic headspace injection unit at 60 °C for 10 min, followed by headspace (100 μL) extraction by a heated syringe (65 °C). Nitrogen was drift gas and carrier gas. Drift gas program flow was 150 mL/min for 30 min. The carrier gas program flow rate was set at 2 mL/min for 2 min, raised to 10 mL/min within 8 min, raised again to 100 mL/min within 10 min, and remained at 100 mL/min for 10 min. The analyte is driven to the ionization chamber in positive ion mode by a 3H ionization source with an activity of 300 MBq.

### 2.6. Sensory Analysis

The sensory properties of black raspberry wines were assessed using a quantitative descriptive analysis (QDA) method by an expert group comprising 15 tasters (9 females and 6 males, ranging in ages from 20 to 44). Panelists were recruited according to their motivation and availability. All subjects gave their informed consent for inclusion before they participated in the study. These participants received at least 100 h of sensory analysis training previously, according to ISO 4121 [15] and ASTM-MNL 13 [16], including methods, techniques and the wine tasting process. The samples were repeated twice for two consecutive weeks to ensure the reliability of the data. The aroma descriptors perceived in black raspberry wines were classified in six classes involving floral, fruity, sweet, green, solvent and global aroma. Samples were served at 18 °C in standard wine-testing glasses according to ISO 3591 [17], labeled with three numbers, and presented to the judges in a random order. The judges were required to grade the samples and rate the intensity of each attribute on the basis of a 9-point scale, where zero represented nonexistence and nine corresponded to the highest intensity. Each wine was evaluated in duplicate. The protocol was approved by the Ethics Committee of Ludong University Institutional Review Board.

### 2.7. Statistical Analysis

For the statistical analysis of volatile compounds, SPSS 22.0 (SPSS Inc., Chicago, IL, USA) was used to conduct one-way analysis of variance (ANOVA) and Duncan’s multi-range test, and model *p* = 0.05 was statistically significant. In addition, principal component analysis (PCA) was used to gain a more comprehensive understanding of volatile compounds determined by HS-GC-IMS and to explore possible correlations between the black raspberry wines tested.

For the elucidation of the relationship between sensory descriptors and volatile compounds, partial least squares regression (PLS) analyses were conducted with SIMCA 13.0 software (Umetrics, Umea, Sweden). All variables were mean-centered and normalized to unit variance before applying PLS analyses.

## 3. Results and Discussion

### 3.1. Fermentation Process

During MLF, the bacterial growth of four MLF starters and the content of L-malic acid in the corresponding wines were determined, and the results were shown in Figure 1 and Figure 2. The growth dynamics of *L. plantarum* Viniflora NoVA and *L. plantarum* NV27 showed similar trends, since they showed prosperous growth shortly after inoculating into the black raspberry wine and had early starts of the degradation of L-malic acid. At the 12th day, NoVA acquired a maximal concentration of around 5.13 × 10^7^ cfu/mL in wine samples, but rapidly decreased after then. In contrast, *O. oeni* O-Mega and *O. oeni* DS04 were slightly inhibited at the start of MLF and a lag in the reduction of L-malic acid occurred, but then those two *Oenococcus* strains showed rapid growths for around 6 days, and sharp decreases in L-malic acid were immediately observed. At full MLF, O-Mega and DS04 had a maximal bacterial population of 7.08 × 10^7^ and 5.62 × 10^7^ cfu/mL in their corresponding wines, and both slightly decreased till the completion of MLF. Similar growth trends of *Lactiplantibacillus* and *Oenococcus* strains were also observed in the fermentation of cherry wine [18].

As for the duration of MLF shown in Figure 2, it was found that *L. plantarum* Viniflora NoVA only took 20 days to finish MLF, and a similar duration time was observed for *L. plantarum* NV27. The use of *O. oeni* O-Mega and *O. oeni* DS04, however, extended the MLF period to 22 and 23 days, respectively, proving that *Lactiplantibacillus* strains exhibited greater fermentation efficiencies over *Oenococcus*, which is in agreement with a few reports conducted in such an aspect [19,20].

### 3.2. Basic Composition

The basic compositions of black raspberry wines fermented individually by four LABs were shown in Table 1. It was seen that these physicochemical parameters were all within the rank of values accepted for regular wines. The total acidity varied from 9.18 to 9.27 g/L among the tested samples, with the highest value in BRL-2 and lowest in BRO-1. The pH was decreased accordingly, and this value in black raspberry wine inoculated with *Lactiplantibacillus* was lower than that with *Oenococcus*, suggested that the *Oenococcus* strains were more capable in the reduction of organic acids. As for residual sugars, the use of *L. plantarum* led to a more complete fermentation, causing a lower content of this parameter. With regard to volatile acidity, in the wines inoculated with *Oenococcus* strain, as evidenced that a respective value of 0.38 and 0.40 g/L of such a parameter, were achieved in BRO-1 and BRO-2, both lower than that from *Lactiplantibacillus* inoculated wine.

### 3.3. Volatile Composition

A headspace GC-IMS technique was used in our study for the identification and quantification of volatile substances resulting from four black raspberry wines. Such a technique has the advantages of simple system configuration, easy sampling, high detection efficiency and low price [21,22], in comparison with the classic HS-SPME-GC-MS methodology. The volatile compounds in our wine samples were visualized by a 3D topographic map (Figure A1), where the *y*-axis represented GC retention time, the *x*-axis represented drift time for identification, and the *z*-axis represented peak height for quantization. A total of 31 typical volatile compounds were identified, including esters, alcohols, ketones, aldehydes and terpenes (Table A1). The schematic diagram of selected signal peak area of typical target compounds was shown in Figure 3, and the mean and standard deviation values of signal strength were listed in Table 2. As seen from the data, the volatile components varied significantly among the tested raspberry wines.

Esters were the most significant volatile substances in black raspberry wine. A total of 15 volatile esters were detected from the four samples, the main representatives being ethyl acetate, ethyl formate, ethyl lactate and ethyl hexanoate. BRO-1 contained higher amounts of methyl 2-methylbutanoate, ethyl lactate, ethyl hexanoate and isoamyl acetate. BRO-2 was characterized by a high abundance of ethyl formate, ethyl lactate, isobutyl acetate, propyl acetate and ethyl 2-methylbutanoate. BRL-1 showed higher intensities of ethyl propanoate, ethyl octanoate and methyl 2-methylbutanoate while BRL-2 had higher levels of methyl 2-methylbutanoate, ethyl hexanoate, isoamyl acetate and ethyl butyrate. Three ethyl esters, including ethyl propanoate, ethyl nonanoate and ethyl hexanoate, showed similar intensities in BRL-1 and BRL-2.

From those data, it was found that *Lactiplantibacillus* facilitated the synthesis of ethyl hexanoate, ethyl propanoate, ethyl octanoate, isoamyl acetate and ethyl 2-methylbutanoate over *Oenococcus* in our study. A study of MLF in Pinot Noir wine, aimed to make a comparison of *Lactiplantibacillus* and *Oenococcus* on the production of volatile compounds, found that the former was superior to the latter producing ethyl propanoate, ethyl hexanoate and isoamyl acetate, which was consistent with our result [7]. Additionally, previous studies had reported the ability of *Oenococcus* to produce ethyl lactate, isobutyl acetate and propyl acetate in wine [23], which was in agreement with our result. Moreover, some important esters, such as ethyl butyrate, isoamyl acetate and ethyl hexanoate, which can contribute desirable and fruity sensory properties to the wine including banana, strawberry and green apple [24,25], had higher intensities in BRL2 wine, suggesting that such a wine could be perceived a more intense fruity aroma.

Eight volatile alcohols were detected in the samples, mainly involving 3-methyl-3-buten -1-ol, 2-methyl-1-propanol, ethanol and 1-propanol. The signal intensity of total alcohol in a decreasing tendency was BRL-2, BRO-2, BRO-1 and BRL-1. BRO-1 was characterized by a high abundance of 1-propanol and 3-methyl-1-pentanol. BRO-2 had higher levels of 2-methyl-1-propanol and 1-butanol. BRL-2 showed a higher signal intensity of 3-methyl- 3-buten-1-ol. The intensity of 1-butanol-D was about 1.30 and 3.24-fold than those in BRL1 and BRL2, respectively. 1-Hexanol differed slightly in these wine samples treated with different LAB. Previous studies have shown that the production of 1-butanol and 1-hexanol in Pinot Noir wine was higher when cultured with *Oenococcus* than that with *Lactiplantibacillus* [7]. A similar finding appeared in the study [3], which also found that *Oenococcus* has a slight advantage over *Lactiplantibacillus* in the production of 2-methyl-1-propanol, 1-propanol and 3-methyl-1-pentanol.

Two aldehydes and four ketones were detected in the samples, mainly including acetone and nonanal. The native LAB seemed to favor the production of this volatile group than their commercial counterparts, as seen that a larger total amount of ketones was detected in BRL2 than in BRL1 and a larger total amount of this volatile group was found in BRO2 than in BRO1. There was no statistical difference in the signal intensity of butanal among the experimental wines. The concentration of 2-octanone in BRO-1 and BRO-2 was similar and around 2-fold higher than those in the other two wines. *Oenococcus* has been found facilitating the production of 2-octanone during wine fermentation [26], and such finding was in agreement with our current result. BRO-2 had the highest abundance of nonanal and 2-pentanone, which was consistent with previous report [21], which found *Oenococcus* was beneficial for the production of nonanal in cider.

In addition to the aroma compounds mentioned above, two other compounds were detected, α-terpinene and terpinolene involved. The former compound was relatively more synthesized by the native *Lactiplantibacillus* starter, while the levels of latter in BRO-2, BRL-1 and BRL-2 were basically identical.

### 3.4. Principal Component Analysis

In order to further understand the effects of different MLF starters on the volatile spectrum of black raspberry wine, a principal component analysis (PCA) was used. The first two principal components can only explain 68.6% of the total variance. Therefore, 11 redundant variables were removed from the matrix whose coefficients <0.7, and a new set with 26 variables (data matrix 12 × 26) accounting for 85.49% of the total variance was acquired (Figure 4). As seen, BRL-1 and BRL-2 were correlated with the variables strongly related to negative PC1 and clustered in the second and third quadrants, showing a good correlation with a variety of volatile esters, including ethyl isobutyrate, ethyl propanoate, methyl 2-methylbutanoate-D, ethyl nonanoate, ethyl hexanoate-M, ethyl hexanoate-D, isoamyl acetate-M. BRO-2 was more affected by parameters strongly related to positive PC1 and PC2, located in the first quadrant and had strong correlation with isobutyl acetate, propyl acetate, 2-pentanone, nonanal, 2-methyl-1-propanol and 1-butanol-D. BRO-1, which was more affected by parameters strongly related to negative PC2, was located in the fourth quadrant and showed a good correlation with 3-methyl-1-pentanol and methyl 2-methylbutanoate-M.

### 3.5. Sensory Analysis

The sensory properties of all tested black raspberry wines were evaluated by an expert group and the QDA results were shown in Figure 5. BRO-1 wine was characterized by the highest intensity of ‘floral’ and relatively low for most of the ratings. Compared to BRO-1, BRO-2 was rated higher in several attributes including ‘fruity’, ‘green’, ‘solvent’ and ‘sweet’. BRL-1 was found to significantly improve the description of ‘fruity’. BRL-2 was characterized by the strongest intensities of most rated attributes, including ‘fruity’, ‘green’, ‘sweet’ and ‘solvent’. Although BRL-2 had a higher attribute intensity of ‘fruity’ and ‘sweet’, the description of ‘solvent’ and ‘green’ also received a higher score, thereby reducing the global aroma quotient accordingly. The overall aroma of the five black raspberry wines were BRL-1, BRO-2, BRL-2, and BRO-1 in the decreasing trend.

### 3.6. Relationship between PLS Analysis of Volatile Compounds and Sensory Description

Partial least squares regression analysis was carried out with SIMCA 16 software to determine the relationship between the volatile components of black raspberry wine and the sensory descriptors. The initial model correlation was weak, and 16 redundant variables were removed, and a new data set with 21 variables was generated. The R^2^X of such new data set was 0.876 (over the threshold of 0.8), R^2^Y reached 0.973, and the difference between R^2^Y and Q^2^Y is 0.179 (below the threshold of 0.3). The values of these parameters indicated that the newly constructed PLS was adequate for the explanation of the relationship between sensory attributes and those volatile components (Figure 6).

As shown, a total of four compounds were found correlated with the ‘floral’ note, including ethyl lactate (lactic, buttery, fruity), 3-methyl-1-pentanol (fusel, medicinal), 2-octanone (earthy, weedy, natural, woody, herbal) and 2-methyl-1-propanol (bitter, green, harsh) [27,28], and the former two compounds had VIP > 1. ‘Fruity’ was associated with ethyl isobutyrate (sweet, ethereal, fruity), isopropyl alcohol (alcohol, musty, woody), terpinolene (fresh, woody, sweet, pine), ethyl propanoate (sweet, fruity, rummy, juicy), methyl 2-methylbutanoate (green, apple), ethyl octanoate (pear), 3-hydroxy-2-butanone (sweet, buttery, creamy) and ethyl hexanoate (pineapple) [29,30], with VIP > 1 for the first three compounds. Ethyl formate (ethereal, alcohol, rose) and 2-pentanone (sweet, fruity, ethereal, wine, banana) were associated with the sensory properties of ‘solvent’. The relationship between the sensory properties of ‘green’ and ‘sweet’ and the components of PLS was similar, mainly related to 3-methyl-3-buten-1-ol (banana, cognac), acetone (solvent, ethereal, apple, pear), butanal (fusel, medicinal), α-terpinene (lemon, herbal, medicinal, citrus), ethyl lactate (lactic, buttery, fruity), ethyl 2-methylbutanoate (apple) [27,29,30], and the VIP of the former four compounds was >1.

By integrating the PLS and sensory evaluation results, it was found that the higher sense of ‘floral’ in BRO1 (with *O. oeni* O-Mega) may be due to a high concentration of ethyl lactate and 3-methyl-1-pentanol in this wine. The stronger ‘solvent’ note appeared in the BRO-2 and BRL-2 black raspberry wine (separately fermented by *O. oeni* DS04 and *L. plantarum* NV27), was possibly associated a higher amount of ethyl formate and 2-pentanone. The use of NV27 also led to a strong fruity aroma, which could be correlated with a higher level of ethyl isobutyrate, isopropyl alcohol and terpinolene in its resulting wine. Finally, higher ‘green’ and ‘sweet’ odors assessed in BRO-2 and BRL-2 may be linked to a higher amount of 3-methyl-3-buten-1-ol, acetone, butanal and α-terpinene in these two wines.

## 4. Conclusions

This study investigated four LAB starters, including two *Oenococcus* and two *Lactiplantibacillus* species, on the production of black raspberry wine. Combined the results from fermentation performance, aroma quantification and sensory evaluation, the native *L. plantarum* NV27 was found presenting the best overall result. In the following study, a large-scale production of wine-making will be conducted. It provided data support for the selection of *Oenococcus* and *Lactiplantibacillus* during the fermentation of black raspberry wine, which was favorable to the selection of strains for the target product style.

## Figures and Tables

**Figure 1 foods-12-04212-f001:**
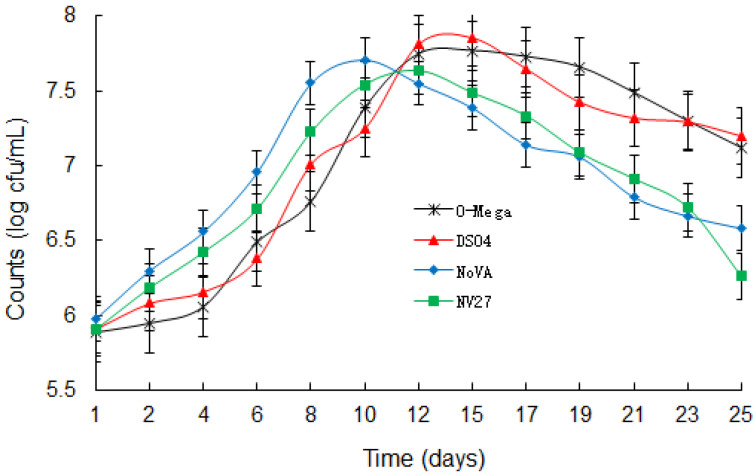
Count changes in different LABs during MLF of black raspberry wine.

**Figure 2 foods-12-04212-f002:**
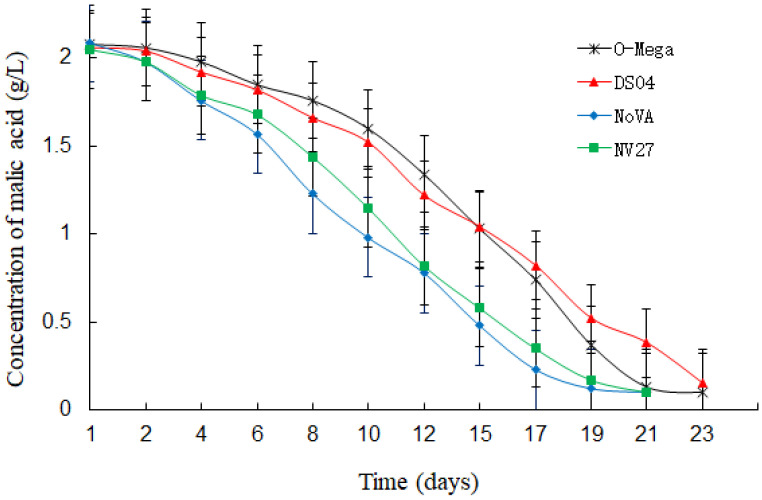
Malic acid degradation of black raspberry wine during MLF by different LABs.

**Figure 3 foods-12-04212-f003:**
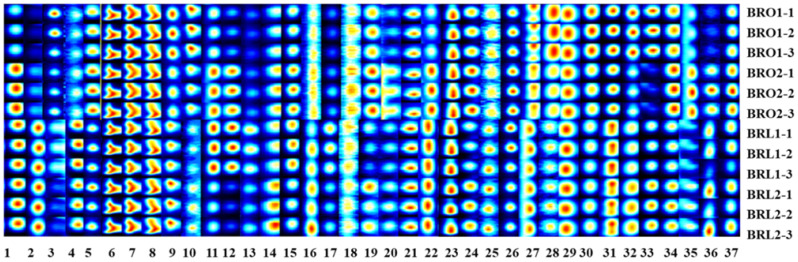
The schematic diagram of selected signal peak area of typical target compounds in black raspberry wines.

**Figure 4 foods-12-04212-f004:**
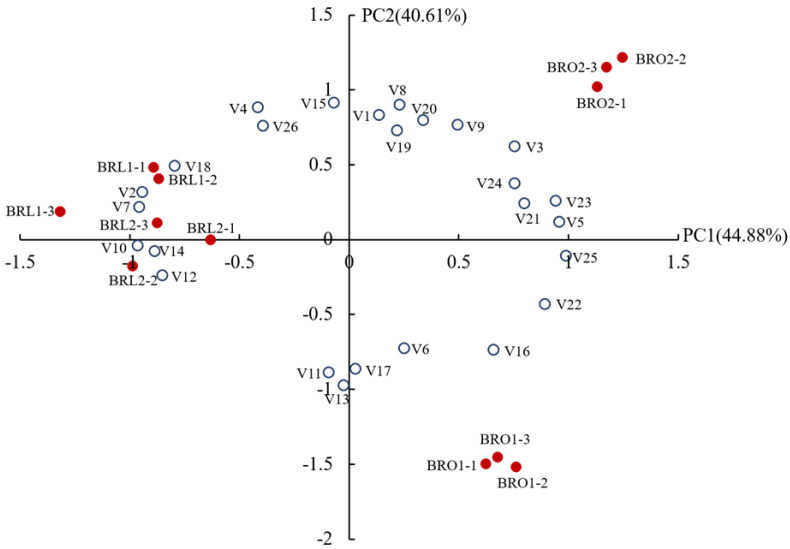
The principal composition analysis of the volatile compounds in the samples treated by 4 LAB. V1: ethyl acetate; V2: ethyl propanoate; V3: propyl acetate; V4: ethyl isobutyrate; V5: isobutyl acetate;V6: methyl 2-methylbutanoate-M; V7: methyl 2-methylbutanoate-D; V8: ethyl 2-methylbutanoate; V9: ethyl 3-methylbutanoate; V10: ethyl hexanoate-M; V11: ethyl hexanoate-D; V12: isoamyl acetate-M; V13: isoamyl acetate-D; V14: ethyl nonanoate; V15: ethyl lactate-M; V16: ethyl lactate-D; V17: 1-propanol; V18: isopropyl alcohol; V19: 1-butanol-M; V20: 1-butanol-D; V21: 2-methyl-1-propanol; V22: 3-methyl-1-pentanol; V23: nonanal; V24: 2-pentanone; V25: 2-octanone; V26: terpinolene.

**Figure 5 foods-12-04212-f005:**
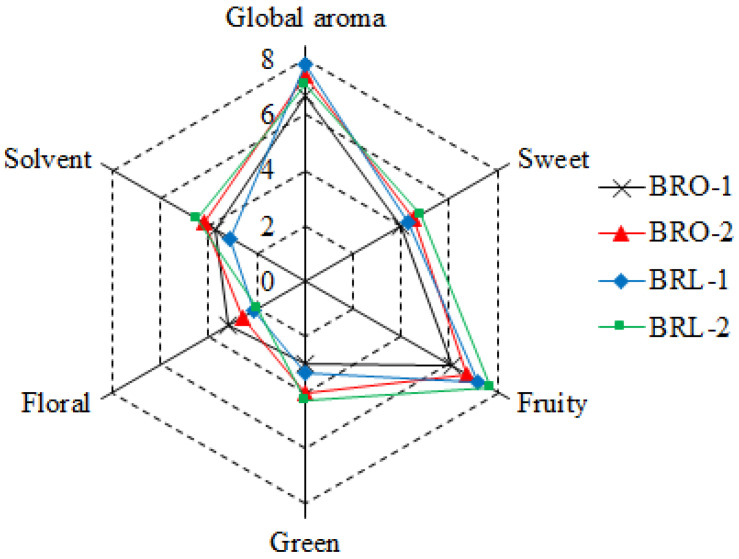
Average values of sensory evaluation scores of black raspberry wines made from different MLFs.

**Figure 6 foods-12-04212-f006:**
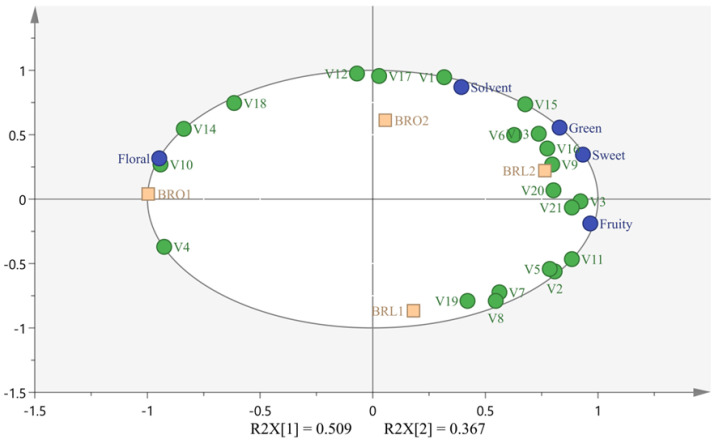
PLS analysis of volatile compounds and sensory description in black raspberry wine. V1: ethyl formate; V2: ethyl propanoate; V3: ethyl isobutyrate; V4: methyl 2-methylbutanoate-M; V5: methyl 2-methylbutanoate-D; V6: ethyl 2-methylbutanoate; V7: ethyl hexanoate-M; V8: ethyl octanoate; V9: ethyl lactate-M; V10: ethyl lactate-D; V11: isopropyl alcohol; V12: 2-methyl-1-propanol; V13: 3-methyl-3-buten-1-ol; V14: 3-methyl-1-pentanol; V15: butanal; V16: acetone; V17: 2-pentanone; V18: 2-octanone; V19: 3-hydroxy-2-butanone; V20: α-terpinene; V21: terpinolene.

**Table 1 foods-12-04212-t001:** Basic compositions of black raspberry wines fermented by different LAB.

Parameters *	After AF	BRO-1	BRO-2	BRL-1	BRL-2
pH	3.44 ± 0.01	3.50 ± 0.01 a	3.51 ± 0.01 a	3.48 ± 0.01 a	3.49 ± 0.01 a
Residual sugar (g/L)	2.88 ± 0.02	2.09 ± 0.02 b	2.24 ± 0.02 c	1.17 ± 0.02 a	1.23 ± 0.03 a
Volatile acidity (g/L)	0.29 ± 0.01	0.38 ± 0.02 a	0.40 ± 0.02 a	0.42 ± 0.03 a	0.47 ± 0.02 b
Total acidity (g/L)	9.61 ± 0.04	9.18 ± 0.05 a	9.22 ± 0.04 a	9.23 ± 0.04 a	9.27 ± 0.04 a
Ethanol (%)	11.03 ± 0.05	10.88 ± 0.04 a	10.74 ± 0.06 a	11.05 ± 0.09 b	11.10 ± 0.10 b
Dry extract (g/L)	34.0 ± 0.5	32.6 ± 0.5 a	31.9 ± 0.5 a	33.0 ± 0.4 ab	33.4 ± 0.5 b

* Values with different superscript roman letters (a–c) in the same row are significantly different according to the Duncan test (*p* = 0.05).

**Table 2 foods-12-04212-t002:** Signal intensities of volatile compounds in black raspberry wine by GC-IMS.

Compounds *	BRO1	BRO2	BRL1	BRL2
Esters				
Ethyl isobutyrate	107.703 ± 3.684 a	221.020 ± 2.033 b	218.183 ± 6.634 b	225.047 ± 9.265 b
Ethyl propanoate	54.393 ± 0.655 a	72.903 ± 0.554 b	158.403 ± 9.462 c	155.623 ± 8.847 c
Methyl 2-methylbutanoate-M	69.780 ± 1.586 d	39.733 ± 2.850 b	52.607 ± 0.663 c	31.940 ± 0.467 a
Methyl 2-methylbutanoate-D	43.010 ± 2.914 a	47.653 ± 1.526 b	121.157 ± 0.670 c	127.740 ± 3.234 d
Ethyl acetate	2641.353 ± 69.427 a	2919.587 ± 22.880 b	2854.257 ± 54.245 b	2695.293 ± 118.110 a
Ethyl formate	587.293 ± 9.184 b	703.973 ± 11.123 c	491.847 ± 17.171a	693.387 ± 35.134 c
Ethyl nonanoate	50.630 ± 5.623 a	43.960 ± 2.511 a	65.497 ± 5.691 b	64.503 ± 2.443 b
Ethyl octanoate	121.980 ± 13.178 a	128.920 ± 2.666 a	150.210 ± 14.800 b	133.793 ± 7.801 ab
Ethyl lactate-M	300.407 ± 11.082 a	386.587 ± 21.771 c	355.847 ± 4.176 b	365.860 ± 8.238 bc
Ethyl lactate-D	114.487 ± 0.503 c	52.843 ± 3.864 b	25.527 ± 0.711 a	26.977 ± 0.365 a
Ethyl hexanoate-M	253.450 ± 7.137 a	241.017 ± 3.186 a	287.283 ± 12.983 b	282.663 ± 10.136 b
Ethyl hexanoate-D	91.013 ± 3.581 c	68.190 ± 3.526 a	75.893 ± 5.853 ab	81.150 ± 6.747 b
Isoamyl acetate-M	194.910 ± 3.321 ab	184.210 ± 0.265 a	203.693 ± 8.404 b	206.877 ± 10.912 b
Isoamyl acetate-D	258.697 ± 3.147 d	98.830 ± 1.408 a	132.530 ± 6.457 b	190.977 ± 13.594 c
Ethyl butyrate	176.967 ± 2.340 b	153.747 ± 1.589 a	144.117 ± 14.776 a	198.247 ± 15.421 c
Isobutyl acetate	86.443 ± 0.090 c	114.173 ± 0.878 d	35.640 ± 2.422 a	58.153 ± 1.804 b
Propyl acetate	35.857 ± 2.198 b	89.043 ± 1.573 c	29.207 ± 1.045 a	36.107 ± 1.909 b
Ethyl 2-methylbutanoate	19.930 ± 1.428 a	103.137 ± 8.491 c	53.307 ± 17.010 b	68.453 ± 20.714 b
Ethyl 3-methylbutanoate	94.653 ± 4.332 a	137.337 ± 3.891 b	105.527 ± 20.149 a	103.977 ± 12.443 a
Alcohols				
3-Methyl-3-buten-1-ol	4074.960 ± 103.636 a	4190.300 ± 54.368 b	4065.073 ± 14.993 a	4398.860 ± 15.909 c
2-Methyl-1-propanol	1856.930 ± 33.980 b	1977.940 ± 28.017 c	1715.123 ± 23.318 a	1859.720 ± 12.170 b
Ethanol	3441.210 ± 74.189 a	3364.883 ± 50.225 a	3436.547 ± 40.510 a	3431.833 ± 27.747 a
1-Propanol	631.563 ± 18.098 c	523.727 ± 5.672 a	525.577 ± 15.288 a	601.093 ± 14.209 b
3-Methyl-1-pentanol	106.010 ± 2.342 d	85.280 ± 4.787 c	36.707 ± 1.208 a	43.230 ± 2.270 b
1-Butanol-M	107.667 ± 2.015 b	162.567 ± 2.987 d	154.570 ± 5.768 c	98.707 ± 0.677 a
1-Butanol-D	37.803 ± 1.406 a	107.743 ± 3.060 c	82.833 ± 5.679 b	33.320 ± 1.761 a
1-Hexanol-M	27.867 ± 2.000 ab	32.983 ± 3.758 b	30.747 ± 3.522 ab	25.017 ± 3.273 a
1-Hexanol-D	15.940 ± 0.251 ab	12.710 ± 3.332 a	15.230 ± 1.725 ab	22.190 ± 8.267 b
Isopropyl alcohol	38.637 ± 1.301 a	47.583 ± 1.480 b	60.227 ± 6.981 c	59.067 ± 3.021 c
Aldehydes and ketones				
Nonanal	106.110 ± 5.987 b	131.587 ± 2.312 c	82.977 ± 4.422 a	81.013 ± 4.422 a
Butanal	17.000 ± 1.215 a	18.747 ± 1.052 a	17.060 ± 1.194 a	18.863 ± 0.621 a
3-Hydroxy-2-butanone	55.507 ± 5.366 a	64.993 ± 7.110 a	89.107 ± 5.986 b	64.387 ± 1.759 a
Acetone	85.797 ± 3.468 a	140.050 ± 3.665 c	96.087 ± 3.477 b	281.280 ± 6.148 d
2-Pentanone	46.833 ± 4.456 b	79.703 ± 5.619 c	23.803 ± 2.437 a	51.867 ± 1.474 b
2-Octanone	54.747 ± 1.898 c	59.953 ± 1.275 c	14.547 ± 3.940 a	23.833 ± 4.162 b
Others				
α-Terpinene	25.633 ± 1.376 a	26.083 ± 4.230 ab	27.047 ± 3.727 ab	31.737 ± 1.812 b
Terpinolene	54.220 ± 1.062 a	83.440 ± 2.881 b	83.707 ± 1.276 b	82.073 ± 1.672 b

* Values with different superscript Roman letters (a–d) in the same row are significantly different according to the Duncan test (*p* = 0.05).

## Data Availability

The data used to support the findings of this study can be made available by the corresponding author upon request.

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
