# Peer review of "The Influence of *Lactiplantibacillus plantarum* and *Oenococcus oeni* Starters on the Volatile and Sensory Properties of Black Raspberry Wine"

_foods, 2023, doi:10.3390/foods12234212_

Round 1

Reviewer 1 Report

Comments and Suggestions for Authors

L 104. it would be beneficial to include details on the specific techniques and instruments employed for each parameter.

L 76. It would be beneficial for the authors to discuss the rationale behind the specific sugar content chosen and its potential impact on the flavor and aroma of the black raspberry wine

L 111. The method employed for the determination of volatile components in black raspberry wine is described in detail, providing valuable insights into the analytical approach. However, a noticeable discrepancy arises regarding the volume of wine sample used for the analysis. In the text, it states that 1 mL of wine sample was used, whereas the reference cited mentions the use of 0.1 mL of wine. This inconsistency should be clarified and rectified to ensure the accuracy of the methodology section.

L119-130. To further strengthen this section, the authors could provide insights into the specific training process of panelists and any calibration steps taken to ensure consistency in sensory assessments.

L143-163: It has come to my attention that there are notable omissions in the presentation of the study's findings and methodology. Firstly, in the graph displaying the bacterial counts of the four MLF starters, there is no mention of the results for malic acid content. Including the data for malic acid is critical, Secondly, the experimental section lacks a description of the method employed for the determination of malic acid content. I recommend that the authors address these issues promptly by either including the malic acid results in the graph or providing a separate figure to represent malic acid concentration over time. Additionally, the experimental section should be updated to describe the method used for malic acid determination, including any relevant instruments, reagents, and conditions.

Comments on the Quality of English Language

The quality of English language in the text is generally good. The text is well-structured and conveys scientific information clearly and professionally. However, there are a few areas where improvements can be made. In some instances, there are minor grammatical issues, such as missing articles ("a" or "an") or tense consistency. Additionally, there are some sentences that could benefit from further clarification and elaboration, particularly in the explanation of methods and their significance. Addressing these minor language issues and providing more context where necessary would enhance the overall quality of the text.

Reviewer 2 Report

Comments and Suggestions for Authors

Dear Authors,

The topic of the manuscript is very interesting and a lot of experimental work was done. However, there are some things that should be clarified in order to improve manuscript quality.

Title: There is new taxonomy of LAB since 2020. According to it Lactobacillus plantarum is changed to Lactiplantibacillus plantarum.

Abstract: It will be good to write MLF in brackets after malolactic fermentation in ln. 1 because afterwards you use MLF in the abstract.

Introduction: ln 14 brewing?

ln 54 brewing?

According to me it is very strange to use brewing because it is something connected to beer, not to wine.

ln 57 implantation rate?

Results and discussion

Figure 1. According to the labels on the Y-axis the figure is only for cells dynamic, not for L-malic acid dynamic

ln 169-171 According to Table 1 there is no significant difference between pH of all the wines produced, so you can remove the statement.

Table 1 It will be better if in Table 1 show the results for wine before MLF and after MLF

Figure 4 It will be easier for understanding if the lines are in different colours

Comments on the Quality of English Language

Minor correction of English is needed.
